# Effects of Caloric Restriction with Protein Supplementation on Plasma Protein Profiles in Middle-Aged Women with Metabolic Syndrome—A Preliminary Open Study

**DOI:** 10.3390/jcm8020195

**Published:** 2019-02-06

**Authors:** Chia-Yu Chang, Yu-Tang Tung, Yen-Kuang Lin, Chen-Chung Liao, Ching-Feng Chiu, Te-Hsuan Tung, Amalina Shabrina, Shih-Yi Huang

**Affiliations:** 1School of Nutrition and Health Sciences, Taipei Medical University, Taipei 110, Taiwan; julia21808@gmail.com (C.-Y.C.); derossi83621@gmail.com (T.-H.T.); amalina.shabrina@yahoo.com (A.S.); 2Graduate Institute of Metabolism and Obesity Sciences, Taipei Medical University, Taipei 110, Taiwan; f91625059@tmu.edu.tw (Y.-T.T.); chiucf@tmu.edu.tw (C.-F.C.); 3Biostatistics Center, Taipei Medical University, Taipei 110, Taiwan; robbinlin@tmu.edu.tw; 4Proteomics Research Center, National Yang-Ming University, Taipei 112, Taiwan; ccliao@ym.edu.tw; 5Center for Reproductive Medicine & Sciences, Taipei Medical University Hospital, Taipei 110, Taiwan

**Keywords:** metabolic syndrome, caloric restriction, protein supplementation, plasma proteomics

## Abstract

Background: Clinical studies have demonstrated that higher protein intake based on caloric restriction (CR) alleviates metabolic abnormalities. However, no study has examined the effects of plasma protein profiles on caloric restriction with protein supplementation (CRPS) in metabolic syndrome (MetS). Therefore, using a proteomic perspective, this pilot study investigated whether CRPS ameliorated metabolic abnormalities associated with MetS in middle-aged women. Methods: Plasma samples of middle-aged women with MetS in CR (*n* = 7) and CRPS (*n* = 6) groups for a 12-week intervention were obtained and their protein profiles were analysed. Briefly, blood samples from qualified participants were drawn before and after the dietary treatment. Anthropometric, clinical, and biochemical variables were measured and correlated with plasma proteomics. Results: In results, we found that body mass index, total body fat, and fasting blood glucose decreased significantly after the interventions but were not different between the CR and CRPS groups. After liquid chromatography–tandem mass spectrometry analysis, the relative plasma levels of alpha-2-macroglobulin (A2M), C4b-binding protein alpha chain (C4BPA), complement C1r subcomponent-like protein (C1RL), complement component C6 (C6), complement component C8 gamma chain (C8G), and vitamin K-dependent protein S (PROS) were significantly different between the CRPS and CR groups. These proteins are involved in inflammation, the immune system, and coagulation responses. Moreover, blood low-density lipoprotein cholesterol levels were significantly and positively correlated with C6 plasma levels in both groups. Conclusions: These findings suggest that CRPS improves inflammatory responses in middle-aged women with MetS. Specific plasma protein expression (i.e., A2M, C4BPA, C1RL, C6, C8G, and PROS) associated with the complement system was highly correlated with fasting blood glucose (FBG), blood lipids (BLs), and body fat.

## 1. Introduction

Metabolic syndrome (MetS) is an escalating global public health challenge. The prevalence of MetS varies among countries and is affected by region, sex, age, and ethnicity [1,2]. According to the clinical diagnostic criteria of the International Diabetes Federation, approximately 25% of adults have MetS [2]. A sedentary lifestyle, high body mass index (BMI), and relatively high socioeconomic status have been associated with MetS. Studies have also reported that genetics, dietary habits, level of physical activity, cigarette smoking, familial history of diabetes mellitus (DM), and level of education influence the prevalence of MetS and its components [3]. Furthermore, according to the Ministry of Health and Welfare in Taiwan, cardiovascular disease, DM, and hypertension associated with MetS are listed in the ten leading causes of death in Taiwan [4].

Clinical studies have shown that caloric restriction (CR) causes weight loss, changes body composition, and decreases a person’s basal metabolic rate [5,6,7]. These effects are believed to result from lowering the thermic effect of feeding and reducing thermal substrates. Thus, it has been considered that CR can cause metabolic adaptation. A study reported that short-term CR influenced the secretion of adipokines from adipocytes (e.g., increased adiponectin levels and decreased leptin levels in blood), but there was no significant difference in the amount of fat-free mass [8]. Such a change has been regarded as in accordance with alleviation of diseases associated with MetS (e.g., atherosclerosis and type 2 DM) [8]. Additionally, CR has been shown to reduce visceral fat [7] and has been highly correlated with improvements in insulin levels in people with obesity. However, the effects of CR for alleviating metabolic abnormalities are usually not as notable as expected. A study reported that caloric restriction with protein supplementation (CRPS) promoted weight loss, improved the biochemical markers associated with metabolism, increased lean body mass preservation, accelerated adipose tissue catabolism, and helped maintain weight loss by increasing satiety, thermogenesis, and myofibrillar protein synthesis [9,10]. However, proteomic animal and human studies have reported that plasma or tissue protein profiles might be influenced by CR [11,12] and may be partly responsible for changes in metabolism.

To the best of our knowledge, no study has examined the effect of plasma protein profiles on CRPS in MetS. Studies have reported that middle-aged women have greater difficulty than men in reducing body weight [13,14]. Therefore, using a proteomic perspective, this pilot study investigated whether CRPS ameliorated metabolic abnormalities associated with MetS in middle-aged women. This study also investigated the effects of CRPS on blood biochemical characteristics (e.g., triglyceride (TG), total cholesterol (TC), low-density lipoprotein cholesterol (LDL-C), high-density lipoprotein cholesterol (HDL-C), interleukin 6 (IL6), haemoglobin A1c (HbA1c), fasting blood glucose (FBG), blood insulin levels, and body composition.

## 2. Materials and Methods

This pilot study was primarily conducted to analyse plasma protein profiles by using plasma samples of middle-aged women with MetS obtained from a cohort study [15,16]. A brief description of the selection criteria and study design follows.

### 2.1. Participants

The cohort study was conducted at Taipei Medical University (TMU), Taipei Medical University Hospital (TMUH), and Wan Fang Hospital. Volunteer recruitment occurred between May 2012 and March 2013. Volunteers were included in the cohort study when they met the inclusion criteria and were diagnosed with MetS in accordance with the modified National Cholesterol Education Program Adult Treatment Panel III [17] and the World Health Organization guidelines [18]. The inclusion and exclusion criteria were as follows [16]: (1) age 30 to 65 years, (2) BMI ≥24 and ≤35 kg/m^2^, and (3) waist circumference (WC) ≥90 cm in men or ≥80 cm in women. Furthermore, participants were included if at least two of the following conditions were met: (1) TG levels ≥150 mg/dL, (2) HDL-C levels <40 mg/dL in men or <50 mg/dL in women, (3) FBG levels >100 mg/dL, or (4) systemic hypertension with systolic blood pressure ≥130 mmHg and diastolic blood pressure ≥85 mmHg. Participants with a history of cardiovascular events, alcohol or substance abuse, or cardiovascular, hepatic, renal, metabolic, endocrine, psychiatric, cerebrovascular, or peripheral vascular diseases were excluded. Participants who were pregnant, taking any type of lipid-lowering, antihypertension, or hypoglycaemic medication were also excluded.

### 2.2. Study Design

The participants in the cohort study were randomly assigned to one of four groups after a 2-week run-in period, and were then asked to follow a 12-week dietary intervention. The four groups were as follows: CR, CRPS, caloric restriction with fish supplementation, and caloric restriction with protein and fish supplementation. During the 2-week run-in period, the participants were consulted by a dietitian to estimate their regular daily dietary intake and basal metabolic rate (Figure 1A).

In this pilot study, we focused on only two of the four groups: CR and CRPS. The inclusion criteria were as follows: female aged ≥40 years, relatively high blood biochemical variable improvements (i.e., BMI, WC, fasting blood glucose, and blood lipids) after the 12-week intervention, adequate plasma sample, and provision of written informed consent. After the participant selection process, seven and six subjects’ plasma samples were selected in CR and CRPS groups for the proteomic analysis, respectively (Figure 1B). This additional study was approved by the Ethics Committee of the Joint Institutional Review Board at TMU (N201704088), registered at ClinicalTrials.gov as NCT01768169, and conducted in accordance with the Declaration of Helsinki.

During the 12-week dietary interventions, the CR group was asked to consume calorie-controlled lunches and dinners prepared by the Department of Nutrition at TMUH. The CRPS group was asked to consume calorie-controlled lunches and Herbalife Formula 1^®^ (Herbalife, Los Angeles, CA, USA) for dinner. Herbalife Formula 1^®^ is a low-calorie nutritional drink (protein powder); the participants were asked to mix 25 g of the protein powder (comprising 11 g of carbohydrates, 0.6 g of fat, and 8 g of protein) with water (81 kcal/serving). Herbalife Formula 1^®^ was provided as a protein supplement to increase the participants’ daily protein intake.

### 2.3. Anthropometry and Analysis of Clinical and Biochemical Variables

The participants’ height, body weight, WC, and body composition were measured at baseline (week 0) and after intervention (week 12). An oral glucose tolerance test and blood drawing were also conducted at both time points. The blood samples were then stored at −80 °C until analysis. Briefly, serum albumin, TG, and TC were measured using an automated analyser (Ortho Clinical Diagnostics VITROS 950, Johnson & Johnson, New Brunswick, NJ, USA). Serum LDL-C and HDL-C were analysed using an automated analyser (Toshiba TBA-c16000, Toshiba, Tokyo, Japan). FBG was determined using an automated analyser (VITOR 5, 1FS, Ortho Clinical Diagnostics, Johnson & Johnson) with Vitros Chemistry Products GLU slides. Serum insulin was analysed using a radioimmunoassay kit (DIAsource, Lovain-La-Nueve, Belgium). HbA1c was determined using an HLC-723 GHb G7 analyser (Tosoh, Tokyo, Japan). Serum C-reactive protein (CRP) concentrations were analysed using an automated analyser (Toshiba TBA-c16000). IL6 was measured using a Qunatikine high-sensitivity commercial enzyme-linked immunosorbent assay kit (R&D Systems, Minneapolis, MN, USA). In this study, the qualified and subject-matched plasma samples were used to analyse the basic biochemical characteristics and protein profile (or proteome) (Table 1 and Table 2).

During the study period, the participants attended ten weekly sessions of a nutrition course that provided information on healthy diets, exercise, dietary habits, and dietary behaviour modification. Participant compliance was assessed as attendance in the weekly sessions.

### 2.4. Liquid Chromatography–Tandem Mass Spectrometry

Five microliter qualified plasma samples were spiked with protein standard β-lactoglobulin (LACB) and were processed with dithiothreitol (DTT) and iodoacetamide (IAA) for reduction and alkylation, respectively. Processed plasma samples were digested with SMART Digest Kit (SMART Digest Trypsin Kit, P/N60109-101, Thermo Fisher, Bedford, MA, USA), reduced, desalted (Millipore® Ziptips Micro-C_18_, P/NZ720003, Sigma, St. Louis, MO, USA), and purified by SOLAµ™ SPE Plates (Thermo Fisher, Bedford, MA, USA). Finally, the processed samples were dissolved in 0.1% formic acid for LC-MS/MS analysis. After sample preparation, digested peptides were loaded into a LTQ-Orbitrap Elite mass spectrometer with a nanoelectrospray ionisation source (Thermo Electron, Waltham, MA, USA) connected to a nanoACQUITY UPLC system (Waters, Milford, MA, USA). Peptide samples were first loaded with a single injection model into the nanoACQUITY UPLC system, and then peptides were captured and desalted on a C_18_ trap column (2 cm × 180 µm, Symmetry C_18_, Waters, Milford, MA, USA). The peptide samples were separated using a BEH130 C_18_ column (25 cm × 75 µm, Waters, Milford, MA, USA) with a 0–95% segmented gradient of 3–40% B for 168 min, 40–95% B for 2 min, and 95% B for 10 min at a flow rate of 0.5 µL/min. Mobile phases were prepared as solution A (0.1% formic acid in water) and solution B (0.1% formic acid in acetonitrile). The eluted peptides were ionised with a spray voltage of 2.33 kV and introduced into the LTQ-Orbitrap Elite mass spectrometer. The mass spectrometer was conducted in the positive ion mode and on the basis of a data-dependent acquisition method (isolation width: 1.5 Da). Mass spectrum data of the peptides were obtained using a full mass spectrometer survey scan (*m*/*z* range of 350–1600) of 30,000. According to the data-dependent acquisition method, the first 15 most intensively charged peptide ions were scanned. High-energy collisional dissociation of the selected precursor peptide ions was stimulated with helium. The MS data were deposited as mzML to the ProteomeXchange Consortium (http://proteomecentral.proteomexchange.org) with identifier PXD012213 (Project DOI: 10.6019/PXD012213). An enzyme-linked immunosorbent assay (ELISA) was conducted to confirm the liquid chromatography–tandem mass spectrometry (LC-MS/MS) proteomic results.

### 2.5. Protein Identification

The acquired proteomic raw data files were then applied to search against a UniProt human protein database 
(containing 162,989 protein sequences; released on April 2017; http://www.uniprot.org/) by using PEAKS 
Studio 7.5 (Bioinformatics Solutions, Waterloo, Ontario, Canada). The settings in PEAKS Studio 7.5 combined with 
UniProt for searching the protein database were as follows: enzyme set as trypsin with a maximum of two missed cleavage 
sites; precursor and fragment mass tolerance of 20 ppm and 0.8 Da, respectively; and false discovery rate <1%, 
obtained through search against a decoy database in all protein and peptide characteristics. A protein was identified 
when at least one unique peptide was matched. Protein quantification was based on label-free quantitative analysis. 
Furthermore, spectrum counts were normalised with the total identified spectra per biological sample and the proteins. 
The proteins (containing at least two matched peptides or one unique peptide) with statistically higher or lower 
peptide counts in the participants (nonparametric Quade’s test was conducted in SAS version 9.4, Cary, NC, USA) 
were considered as different expressions. All mass data of this study have been documented as raw files and peak lists 
in ProteomeXchange. The selected proteins were based on the biochemical characteristics improvements (included: blood 
pressure-, coagulation-, complement system-, glucose metabolism-, inflammatory, lean body mass- and lipid 
metabolism-associated proteins) and missing values in nanoLC-MS/MS based proteomics dataset (Figure 2).

### 2.6. ELISA Analysis of Selected Protein

Commercial available plasma C4b-binding protein (C4BP), complement component C6 (C6), complement component C8 gamma chain (C8G), and vitamin K–dependent protein S (PROS) were respectively measured using the following commercial ELISA kits: (1) C4BP ELISA kit (#EC2202-1, Assaypro, St. Charles, MO, USA); (2) C6 ELISA kit (#EC6101-1, Assaypro, St. Charles, MO, USA); (3) C8G ELISA kit (#EC8120-1, Assaypro, St. Charles, MO, USA); and (4) PROS ELISA kit (#AB190808, Abcam, Cambridge, UK). Seven and six subject-matched plasma samples were used in CR and CRPS groups for the ELISA analysis, respectively.

### 2.7. Statistical Analysis

Differences between the postintervention clinical and biochemical characteristics of the treatment groups were compared using the Mann–Whitney U test. A comparison between the baseline and postintervention clinical and biochemical measurements between the groups was conducted using the Wilcoxon signed-rank test. Using the LC-MS/MS-derived proteomics data, the nonparametric Quade’s test was adopted to compare the different postintervention protein expressions between the treatment groups with baseline measurements as covariates. Moreover, a Spearman’s rank correlation coefficient was calculated to evaluate the relationship between the specific plasma protein expressions and clinical variables. All statistical analyses were performed using SAS version 9.4. Data are presented as the median (75th percentile values in parentheses), and *p* < 0.05 was considered statistically significant.

## 3. Results

### 3.1. Anthropometric and Clinical Characteristics

Among those in the CR (*n* = 44) and CRPS (*n* = 45) groups (Figure 1 and Figure 2), there were seven and six middle-aged female participants, respectively, enrolled in this pilot study. Their plasma samples were processed, and their protein profiles were analysed using LC-MS/MS. According to the database from the cohort study, the groups showed no significant differences in age: CR = 61.97 years and CRPS = 55.84 years, *p* = 0.52. Body weight, BMI, android fat, gynoid fat, total body fat (TBF), and FBG decreased significantly in the groups throughout the 12-week interventions, but no significant differences were observed between the groups (Table 1). The basic characteristics of subjects were shown in Appendix A.

### 3.2. LC-MS/MS Proteomic Analysis

A proteomic analysis conducted using LC-MS/MS (Thermo Scientific LTQ-Orbitrap Elite, Thermo Fisher, Bedford, MA, USA) revealed that postintervention alpha-2-macroglobulin (A2M) protein expression in the CRPS group was significantly higher than in the CR group. Furthermore, C4BPA, complement C1r subcomponent–like protein (C1RL), C6, C8G, and PROS expression was significantly lower in the CRPS group than in the CR group after intervention (Table 2). To verify the results derived from LC-MS/MS, ELISA kits were used to determine the absolute amount of proteins. C4BPA, C6, C8G, and PROS expression between the groups was consistent with the results obtained from LC-MS/MS (Figure 3).

### 3.3. Correlation of Specific Protein Candidates with Biochemical Variables

A Spearman’s rank correlation coefficient was calculated for the specific protein candidates and showed significant differences between the groups with the selected anthropometric and biochemical variables. A significantly positive correlation was observed between C6 and blood LDL-C levels in the CR group (rs = 0.93, *p* < 0.01) (Figure 4A). Furthermore, C1RL plasma levels were negatively correlated with the android-to-gynoid ratio (A/G%) (rs = −0.90, *p* = 0.01) and TBF (rs = −0.84, *p* = 0.04) in the CRPS group. C1RL expression was also negatively correlated with HbA1c levels in the CRPS group (rs = 0.81, *p* = 0.05). C6 levels were positively correlated with TC (rs = 0.94, *p* < 0.01) and LDL-C (rs = 0.99, *p* < 0.01) levels, whereas there was a significantly positive correlation between C8 and FBG levels in the CRPS group (rs = 0.99, *p* < 0.01) (Figure 4B).

### 3.4. Plasma Proteomic Measurements

Figure 5 displays a volcano plot of the plasma selected proteomic ratio measurements within each group throughout the trial. The proteins depicted as red dots represent the plasma protein levels that increased significantly (*p* < 0.05) with a 1.25-fold change (25% change threshold), whereas the proteins depicted as blue dots represent the plasma protein levels that decreased significantly to 25% changes of the original value. Apolipoprotein A2 (APOA2), complement component 3 (C3), A2M, fibrinogen alpha chain, alpha-1-microglobulin/bikunin precursor (AMBP), and fibronectin (FINC) levels decreased significantly following the 12-week intervention in the CR group. Moreover, apolipoprotein B-100 (APOB-100), apolipoprotein C2 (APOC2), and C6 levels increased significantly throughout the intervention in the CR group (Figure 5). Moreover, FINC levels decreased significantly following the 12-week intervention in the CRPS group (Figure 6). The detailed proteins characteristics were shown in Appendix A.

## 4. Discussion

### 4.1. Effects of the Dietary Intervention on the Clinical Variables

After comparing the anthropometric and biochemical variables between the groups, we found no significant differences in baseline measurements. According to the literature review, we found that the short dietary interventions (12 weeks), loose target for the CR group (1500 kcal/per day), and relatively low protein supplementation dose provided (17% kcal/per day) in this study might explain the reasons for the minor effects of CRPS. A study by Flechtner-Mors et al. showed that a relatively high protein intake (35% kcal/day) combined with 1300 kcal/per day contributed to a significant body weight loss throughout a 12-week intervention [19]. Furthermore, Josse et al. suggested that a relatively high protein intake (28% kcal/day) combined with 1500 kcal/per day contributed to significant body weight and body fat mass losses and a significant increase in lean body mass after a 16-week dietary intervention [10]. Further study on protein supplementation should be considered to enable the use of a relatively high protein dose for a longer intervention period.

### 4.2. Effect of Protein Supplementation on Plasma Protein Profiles

The results showed that eight plasma proteins (A2M, C4BPA, C1RL, C6, C8G, and PROS) were significantly different between the groups. Those plasma protein candidates have been associated with inflammation (A2M and C4BPA), the complement system (C1RL, C6, and C8), and coagulation responses (PROS).

A2M is regarded as a powerful protease inhibitor that is involved in proteolysis [20]. Studies have also shown that A2M is an acute phase protein (APP) that increases slightly during inflammatory responses (e.g., surgical trauma, myocardial infarction, and severe burns) in humans [21,22,23]; however, A2M levels are significantly decreased in people who are overweight or obese. Moreover, a significantly negative correlation was found between A2M levels and BMI [24]. A study reported that the expression of A2M is likely to increase under a relatively low consumption of calories and animal-derived proteins [21]. In the current study, A2M plasma levels were significantly higher in the CRPS group than in the CR group after intervention. However, there were no significant differences in IL-6 or CRP levels between the groups throughout the interventions. Thus, we infer that the increased A2M plasma levels in the CRPS group were influenced by the plant-derived protein supplementation.

C4BP has been shown to be involved in not only the complement system but also inflammation. Moreover, it is regarded as a positive APP. Human C4BP is mainly composed of an α chain (C4BPA) and a β chain (C4BPB), and there are three isoforms (α_7_β_1_, α_6_β_1_, and α_7_β_0_) of C4BP in humans [25]. A study suggested that C4BP α_7_β_0_ isoform levels increase significantly in inflammation [26]. Clinical research also showed a significant increase in C4BPA expression in inflammatory responses [27]. We observed a significant decrease in C4BPA protein expression in the CRPS group, which shows the potential benefit of CRPS in inflammatory responses in people with obesity.

C1RL has been associated with the classical pathway and has a catalytic effect on proteolysis of pro-C1s, stimulating the activation of the pathway [28]. Furthermore, animal experiments have demonstrated a significant increase in mRNA C1RL levels under inflammation conditions [29]. C6 and C8G are thought to be part of the membrane attack complex (MAC), a structure typically formed from the activation of a host’s complement system [30]. The MAC is involved in cell lysis, and recent studies have demonstrated that it is also an inflammatory trigger [31,32]. Moreover, studies have suggested that the MAC stimulates the release of proinflammatory substrates following the activation of endothelial cells and platelets [33]. In the current study, we found that C1RL, C6, and C8G plasma levels were significantly lower in the CRPS group compared with the CR group, which suggests that CRPS reduces inflammatory responses in people with obesity.

Studies have reported that PROS is an anticoagulant protein and is mainly involved in catalysing the inactivation process of factor V and factor VIII through activated protein C (APC) [33,34]. A study showed that PROS expression was inhibited by NF-κB in inflammation [35]. However, in the current study, postintervention PROS plasma levels were significantly lower in the CRPS group compared with the CR group. Moreover, IL-6 and CRP levels, which have been associated with inflammatory responses, were not significantly different between the groups. According to the absolute amount of PROS derived from ELISA, the mean plasma level of PROS in the CRPS group was 20.34 µg/mL, which was still within the normal range (20–25 µg/mL) [36].

### 4.3. Spearman’s Correlation of Selected Plasma Protein Candidates with Biochemical Variables

Although we observed several significant differences between the plasma protein candidates and the selected biochemical variables, C6 plasma levels were significantly correlated with LDL-C levels in both groups. A study showed that LDL-C plasma levels decreased through an LDL apheresis (LDL-A) operation in patients with familial hypercholesterolaemia. Moreover, the LDL-A operation was shown to prevent cardiovascular events [37]. In another proteomic study, Yuasa et al. suggested that C6 is involved in the cause and pathophysiology of atherosclerosis [38]. To the best of our knowledge, the current study is the first to report a highly significant difference between C6 and LDL-C levels. Further studies with a larger sample size are required in the future.

### 4.4. Effects of CRPS on Plasma Protein Profiles

FINC plasma levels decreased significantly in both the CR and CRPS groups after the 12-week dietary interventions. A significantly positive correlation was found between FINC plasma levels and the degree of obesity in the participants [39]. Furthermore, the adipose tissue gene expression of FINC was significantly downregulated in those who successfully maintained their weight after the CR intervention and the following weight maintenance phase [40]. Studies have suggested that increased FINC plasma levels are related to DM, cardiovascular disease, kidney disease, and pre-eclampsia [41,42,43,44,45,46]; therefore, the significant decrease in FINC plasma levels throughout the interventions in both groups suggests that CRPS is beneficial in metabolic abnormalities.

### 4.5. Limitations

This study has several limitations. First, although this study was designed as a pilot study, a small sample size led to inadequate statistical power. Second, an inadequate dietary intervention might have influenced the effect of the protein supplementation. Large double-blinded randomised studies on patients with MetS are required to confirm our findings.

## 5. Conclusions

The findings suggest that CRPS improves inflammatory responses in middle-aged women with MetS. Furthermore, specific plasma protein expression associated with the complement system was highly correlated with FBG, blood lipids (BLs), and body fat. The precise effect of CRPS and the relevance of specific plasma proteins as potential predictors of biochemical variables should be investigated in large clinical trials.

## Figures and Tables

**Figure 1 jcm-08-00195-f001:**
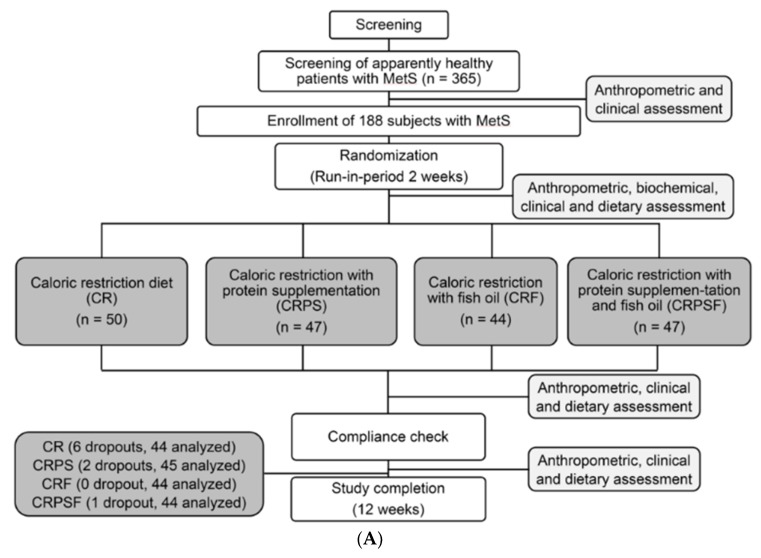
Flow chart of participant selection. (**A**) Participants were randomly assigned to one of four groups: caloric restriction (CR), caloric restriction with protein supplementation (CRPS), caloric restriction with fish oil (CRF), or caloric restriction with protein supplementation and fish oil (CRPSF). (**B**) We focused on only two groups in the current study: CR and CRPS. The participants’ plasma was digested into peptides and then analysed using LC-MS/MS. BMI: body mass index, WC: waist circumference, FBG: fasting blood glucose, BLs: blood lipids, MetS: Metabolic syndrome.

**Figure 2 jcm-08-00195-f002:**
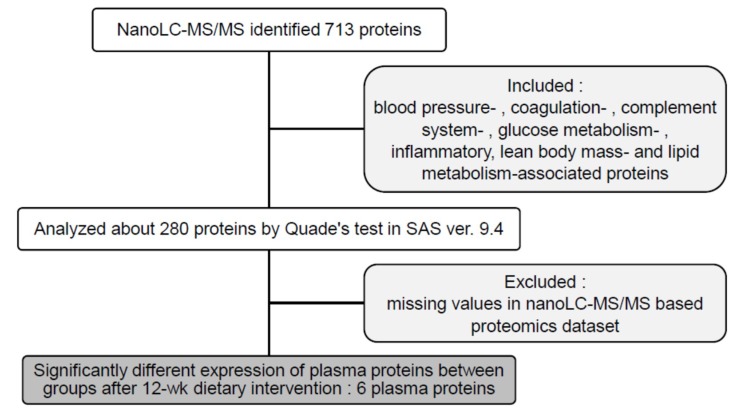
Flow chart of statistical analysis of plasma protein profiles. After the plasma samples (digested peptides) were analysed using nano-LC-MS/MS, PEAKS Studio 7.5 was used to identify and quantify the proteins. nano-LC-MS/MS: nanoflow liquid chromatography–mass spectrometry. 12-wk: 12-week, ver.: version.

**Figure 3 jcm-08-00195-f003:**
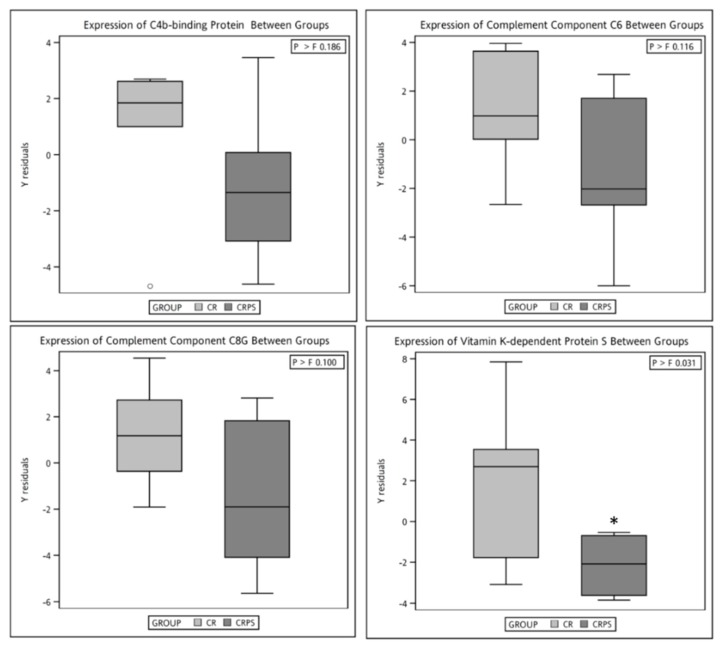
Absolute changes in the plasma levels of several proteins in female patients with metabolic syndrome after the 12-week dietary interventions analysed using an ELISA kit: caloric restriction (CR) or caloric restriction with protein supplementation (CRPS). Box plots represent values between the 25th and 75th percentile. The solid line within the box is the median value, and the symbol within the box is the mean value. The circles are values that are 1.5 times the interquartile range above the upper quartile and below the lower quartile. Units of raw data are ng/mL. * *p* < 0.05 in comparison with postintervention measurements between the groups and according to the Quade’s test with baseline measurements as covariates.

**Figure 4 jcm-08-00195-f004:**
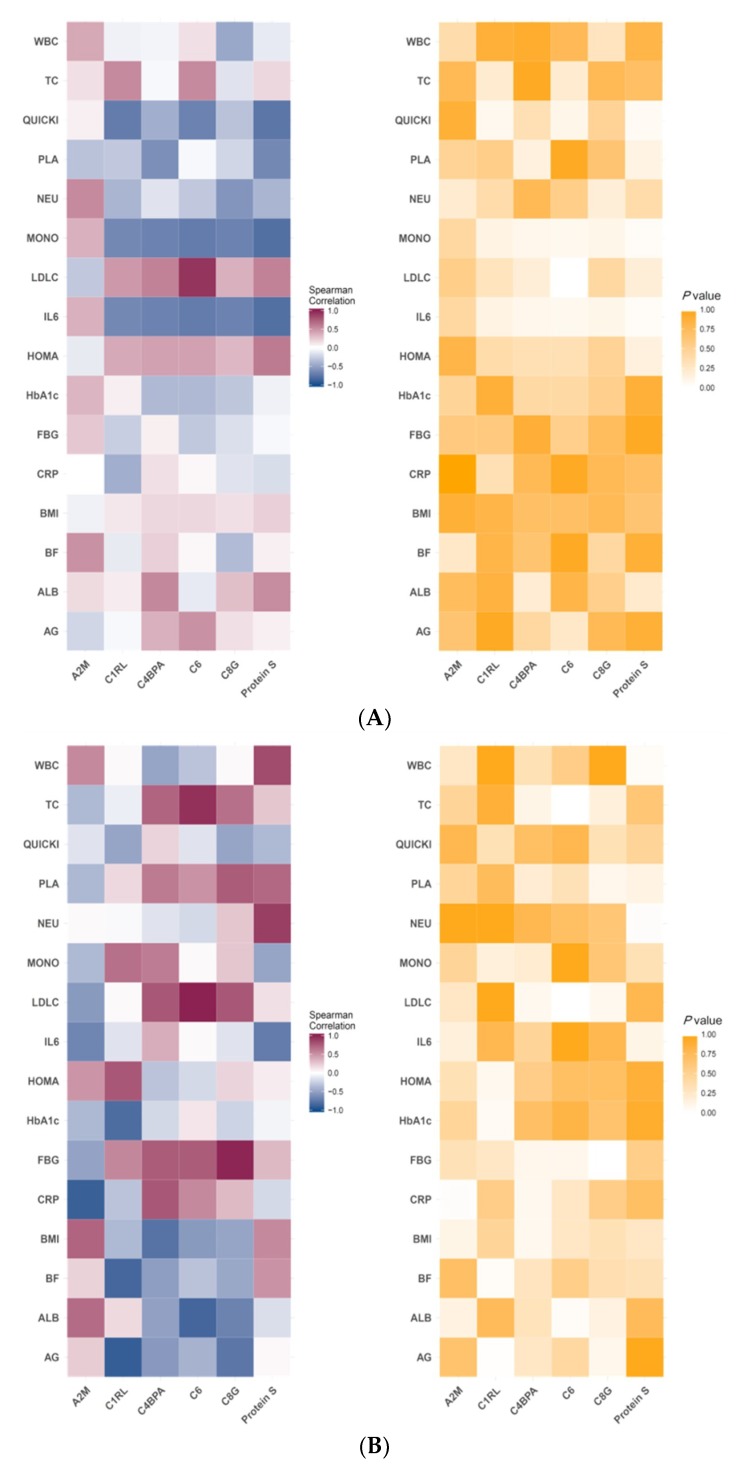
Heat maps showing different baseline and postintervention protein plasma levels. A significant correlation was observed with the selected increment clinical variables between baseline and postintervention in the CR (**A**) and CRPS (**B**) groups. The left red–blue scale heatmaps show the estimates of Spearman’s correlation coefficients, whereas the right yellow-scale heat maps show the *p*-values for the same protein–clinical variable pairs. AG: android/gynoid fat ratio, ALB: albumin, TBF: total body fat, BMI: body mass index, CRP: C-reactive protein, FBG: fasting blood glucose, HOMA: homeostatic model assessment for insulin resistance, IL6: interleukin 6, LDL-C: low-density lipoprotein cholesterol, MONO: mononuclear, NEU: neutrophil, PLA: platelet, QUICKI: quantitative insulin sensitivity check index, TC: total cholesterol, WBC: white blood cell, A2M: alpha-2-macroglobulin, C1RL: Complement C1r subcomponent–like protein, C4BPA: C4b-binding protein alpha chain, C6: complement component C6, C8G: complement component C8 gamma chain, Protein S: vitamin K-dependent protein S, CR: caloric restriction, CRPS: caloric restriction with protein supplementation.

**Figure 5 jcm-08-00195-f005:**
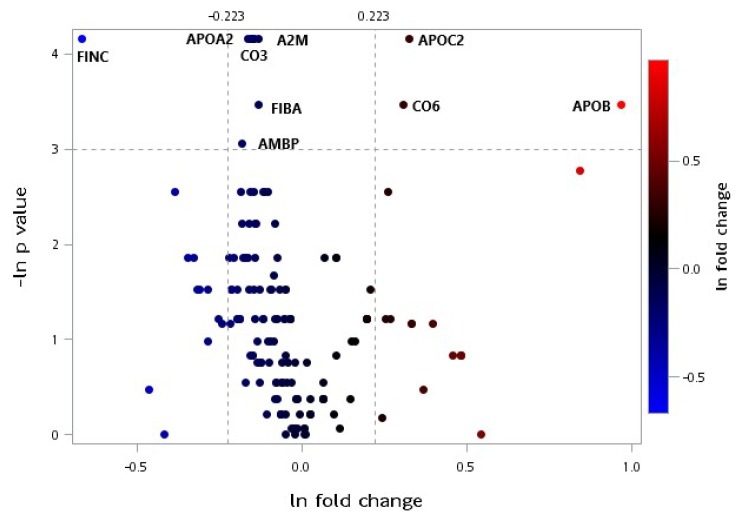
Volcano plot of plasma selected proteomic ratio measurements between baseline and postintervention in the CR group. Fold change means the postintervention value divided by the baseline value, and the *p*-value was according to a Wilcoxon signed-rank test that compared the different protein expressions before and after the interventions within the groups. Thresholds are presented as dotted lines. The fold change cut-off points were 1.25 and 1/1.25, and the *p*-value cut-off point was 0.05. Both data were converted to a natural logarithm, as shown. CR: caloric restriction, A2M: alpha-2-macroglobulin, AMBP: alpha-1-microglobulin/bikunin precursor, APOA2: apolipoprotein A2, APOB: apolipoprotein B-100, APOC2: apolipoprotein C2, CO3: complement component 3, CO6: complement component 6, FIBA: fibrinogen alpha chain, FINC: fibronectin.

**Figure 6 jcm-08-00195-f006:**
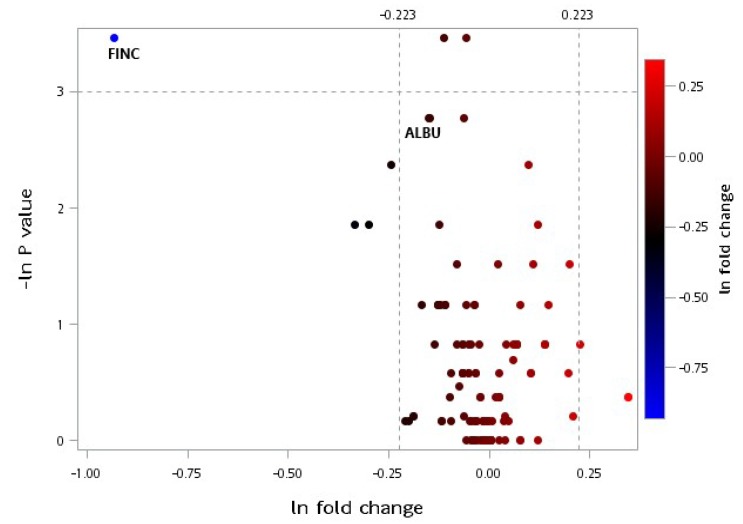
Volcano plot of plasma selected proteomic ratio measurements between baseline and postintervention in the CRPS group. Fold change represents the postintervention value divided by the baseline value, and the *p*-value is associated with a Wilcoxon signed-rank test that compared the different protein expression levels before and after the interventions within the group. Thresholds are presented as dotted lines. The fold change cut-off points were 1.25 and 1/1.25, and the *p*-value cut-off point was 0.05. Both sets of data were converted to a natural logarithm as shown. CRPS: caloric restriction with protein supplementation, ALBU: albumin, FINC: fibronectin.

**Table 1 jcm-08-00195-t001:** Participants’ clinical and biochemical characteristics before and after the 12-week dietary interventions ^a^.

Variable	CR (*n* = 7)		CRPS (*n* = 6)		Post-Intervention
	Baseline	12-Week	Baseline	WK12	*p*-Value ^b^
**Age (y/o)**	61.97 (64.61)	-	55.84 (63.87)	-	0.52
**Anthropometrics**					
BW (kg)	66.10 (75.40)	59.80 (68.90) *	67.90 (77.35)	60.60 (72.23) *	0.94
BMI (kg/m^2^)	28.21 (31.59)	25.46 (28.40) *	27.03 (28.25)	24.50 (26.09) *	0.62
WC (cm)	85.00 (94.00)	79.00 (86.00) *	85.50 (93.50)	76.00 (83.75) *	0.43
**Body Composition**					
Android (%)	53.50 (55.00)	46.30 (53.10) *	47.95 (51.83)	44.20 (48.65) *	0.52
Gynoid (%)	48.40 (49.30)	43.30 (45.90) *	43.60 (49.05)	39.65 (43.80) *	0.23
TBF (%)	43.20 (46.90)	40.00 (44.10) *	40.45 (43.20)	34.85 (40.93) *	0.35
**Blood Pressure**					
SBP (mmHg)	144.0 (162.0)	116.0 (149.0) *	147.5 (152.3)	123.0 (130.0) *	1.00
DBP (mmHg)	84.00 (93.00)	75.00 (86.00) *	81.50 (85.25)	70.50 (76.50)	0.32
**Laboratory Data**					
GOT (U/L)	29.00 (33.00)	24.00 (31.00)	22.00 (26.25)	22.50 (25.50)	0.22
GPT (U/L)	36.00 (42.00)	24.00 (27.00) *	16.50 (21.00)	18.00 (21.50)	0.20
BUN (mg/dL)	15.00 (16.00)	12.90 (19.40)	13.50 (16.35)	13.05 (16.93)	0.62
CR (mg/dL)	0.72 (0.73)	0.64 (0.75)	0.63 (0.67)	0.62 (0.69)	0.28
ALB (g/dL)	4.20 (4.50)	4.40 (4.60)	4.30 (4.30)	4.25 (4.33)	0.07
**Glucose Metabolism**					
FBG (mg/dL)	103.0 (105.0)	97.0 (101.0) *	102.5 (127.0)	91.0 (97.8) *	0.06
PC (mg/dL)	132.0 (155.0)	121.0 (138.0)	146.5 (198.8)	119.5 (135.0)	0.43
Insulin (μIU/mL)	11.00 (14.70)	7.86 (14.40)	10.09 (12.58)	7.43 (9.16) *	0.28
HbA1c (%)	5.70 (5.90)	5.60 (5.90)	5.75 (6.55)	5.70 (6.48)	0.94
HOMA-IR	2.85 (3.70)	1.86 (3.45)	2.60 (3.58)	1.60 (2.07) *	0.18
QUICKI	0.33 (0.33)	0.35 (0.36)	0.33 (0.34)	0.36 (0.37) *	0.22
**Blood Lipids**					
TG (mg/dL)	167.0 (184.0)	100.0 (137.0) *	122.0 (155.3)	96.0 (108.3)	0.78
TC (mg/dL)	209.0 (233.0)	184.0 (207.0)	202.0 (210.3)	180.5 (217.0)	1.00
HDL-C (mg/dL)	49.40 (58.00)	44.70 (54.90)	45.25 (63.50)	41.95 (73.13)	0.78
LDL-C (mg/dL)	129.0 (150.0)	127.0 (138.0)	117.0 (138.8)	117.0 (124.5)	0.52
**Inflammatory Status**					
CRP (mg/dL)	0.36 (0.77)	0.22 (0.58)	0.22 (0.51)	0.21 (0.30)	0.83
IL-6 (pg/mL)	2.42 (3.80)	1.71 (2.59) *	3.50 (4.96)	1.68 (2.86)	0.89

BW: body weight, BMI: body mass index, WC: waist circumference, TBF: total body fat, SBP: systolic blood pressure, DBP: diastolic blood pressure, GOT: glutamate oxaloacetate transaminase, GPT: glutamate pyruvate transaminase, BUN: blood urea nitrogen, CR: creatinine, ALB: albumin, FBG: fasting blood glucose, PC: postprandial glucose, HOMA-IR: homeostatic model assessment for insulin resistance, QUICKI: quantitative insulin sensitivity check index, TG: triglyceride, TC: total cholesterol, HDL-C: high-density lipoprotein cholesterol, LDL-C: low-density lipoprotein cholesterol, CRP: C-reactive protein, IL-6: interleukin 6, CR: caloric restriction diet, CRPS: caloric restriction with protein supplementation. ^a^ Data are presented as median (75th percentile values are in parentheses). ^b^
*p*-value according to Mann–Whitney U test for postintervention-comparing between groups. * *p* < 0.05, in comparison with baseline measurements within groups and according to Wilcoxon signed-rank test.

**Table 2 jcm-08-00195-t002:** Different protein plasma levels between the groups after the 12-week dietary interventions ^a^.

Accession	Entry Name	Protein Name	CR (*n* = 7)	CRPS (*n* = 6)	*p*-Value ^b^
P01023	A2MG_HUMAN	Alpha-2-macroglobulin (A2M)	92.0 (94.1)	105.0 (116.0)	0.01
P04003	C4BPA_HUMAN	C4b-binding protein alpha chain (C4BPA)	7.55 (8.71)	4.19 (6.45)	0.04
Q9NZP8	C1RL_HUMAN	Complement C1r subcomponent-like protein (C1RL)	1.74 (3.05)	0.80 (1.79)	0.02
P13671	CO6_HUMAN	Complement component C6 (C6)	8.22 (9.94)	5.39 (7.50)	0.02
P07360	CO8G_HUMAN	Complement component C8 gamma chain (C8)	1.59 (2.17)	0.20 (1.26)	0.04
P07225	PROS_HUMAN	Vitamin K-dependent protein S (PROS)	4.73 (5.62)	3.50 (4.48)	0.02

CR: caloric restriction diet, CRPS: caloric restriction with protein supplementation. ^a^ Data are presented as median (75th percentile values are in parentheses). Units of raw data are spectral counting (SPC)/total spectral counting (T-SPC) × 10,000. ^b^
*p*-value according to Quade’s test for postintervention-comparing between groups with baseline measurements as covariates.

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
