# Peer review of "Effects of Caloric Restriction with Protein Supplementation on Plasma Protein Profiles in Middle-Aged Women with Metabolic Syndrome—A Preliminary Open Study"

_jcm, 2019, doi:10.3390/jcm8020195_

Round 1
Reviewer 1 Report
Authors describe plasma changes in middle-aged women with metabolic syndrome in response to carloric restriction with protein supplementation. Overall, the study intent is good. However, there were some deficiencies in the way samples were analyzed and results were presented that needs to be corrected. Further, authors didn't observe any statistically significant changes in plasma proteins, which negated the value of the study.
Major concerns:
Eventhough there are >40 patients in each group, clinical and biochemical characteristics were measured in only 7 (CR) and 6 (CRPS) patients. This is problematic. It's not possible to assume that rest of the cohort (i.e. 80% of patients in each group) will follow the same trend.It's unclear how many patients were used for LC-MS/MS study.
It's also not clear whether same patients that had the clinical and biochemical work up performed were utilized for LC-MS/MS. I presume that they were the same. But, the text doesn't make it clear.
It's unclear as to how much protein from each sample was subjected to LC-MS/MS. Were the same amount of protein from each sample taken forward or was the same amount of serum that was utilized for LC-MS/MS? This has a large bearing on interpretation of differences.
Only one of the protein presented in table 2 was validated by ELISA (Figure 4). This brings into question the quality of the LC-MS/MS results.
It's unclear how many samples were utilized for ELISA validation in Figure 4.
Figures 6 and 7 suggest that authors seem to draw a fold change threshold of 0.1 to detect significant differences on top of the p-value. A fold change of 0.1 is not possible to detect accurately with label-free, spectral-counting, based LC-MS/MS method.The average CV of a good label-free experiment of replicates is 25%. Overall, the changes shown in Figures 6 and 7 are very small to have been accurately detected by LC-MS/MS. This is also the main reason why ELISA failed to validate all of the LC-MS/MS candidates that were tried.
It's unclear how the correlations were computed (Figure 5 and 6). Were both pre- and post- values used to compute the correlations? Or were delta pre vs. post values used for correlations? Also, were the protein values derived from LC-MS/MS data or ELISA data?
Authors claim that
Figures 6 and 7 show "protein quantitative ratio" measurements is not
accurate. Label-free LC-MS/MS method is not quantitative.
Minor suggestions:
Figure 3 is not needed.
Word "spectural" in the Table 2 legend should be "spectral".
Normalized spectral count values shown in Table 2 are hard to interpret. A fold change would be easier.
Reviewer 2 Report
- n value could be included in the abstract/background
- Please provide the ProteomeXchange Index number for readers
Page 4, Line 145,146: The statement " ELISA was conducted to verify the liquid chromatography-tandem mass spectrometry (LC-MS/MS) proteomics results" is misleading.
-Sample preparation is not provided in the manuscript.
Reviewer 3 Report
In this manuscript Chang et al. reports mass spectrometry based proteomics profiling of plasma to identify molecular markers distinguishing impact of caloric restriction with (CRPS) or without (CR) protein supplementation in middle aged women diagnosed with metabolic syndrome. The clinical study is very well designed and clinical variables, inclusion and exclusion criteria are adequately described. The study finds 6 proteins to be significantly different between the two study groups and suggest mechanism differentiating the two cohorts. While, the clinical arm of the study is well documented and provides robust data of interest, the analytical pipeline lacks significant details to reproduce the results. The 6 proteins identified by discovery proteomics experiments do not appear strong markers as the verification study using ELISA only finds one of the 6 markers to be significant. With no validation data from an independent cohort and inadequate verification of the markers in the same cohort using an orthogonal approach, there is a big question mark on the proposed markers of interest. In summary, this manuscript although of importance does not provide enough merit to be considered for publication in its current shape. However, the manuscript can be considered for publication if the authors can address following comments: Main Comments (a) Plasma proteomics workflow: Sample preparation of plasma for proteomics is completely missing. Describe how blood was collected and plasma component isolated. How was the sample handled from collection to the start of proteomics preparation? How much plasma sample was taken up for proteomics sample preparation? Did the proteomics preparation employ any kind of depletion methods to alleviate problem of high abundance proteins in plasma? How much sample was injected per sample for mass spectrometry data acquisition? Were there any technical replicates? Provide enough details for readers to replicate the experiment. (b) Verify candidate markers using an orthogonal approach (for example targeted proteomics, western blots) or validate the markers in a new cohort. What is the rationale of presenting ELISA results when only 1 of the 4 tested markers show differential expression? A second concern is that the ELISA verification was performed on only 4 of the 6 markers. Explain. Minor Comments (a) Page 4, line 149: The authors mention using human protein database containing 162,989 protein sequences. Does it include complete human uniprot database with unreviewed sequences? Why did not authors use smaller but completely annotated reviewed protein sequences and their isoform database? Line 153, 1% FDR was applied at protein level, peptide level or PSM level? (b) Figure 2: Describe how the 713 identified proteins were trimmed to 280 proteins for statistical test. Did the authors use GO terms associated with the categories listed in the flowchart (blood pressure, coagulation etc.). Why did the authors select these categories? In a discovery proteomics approach, it doesn’t appear prudent to filter out more than half of the data-set because of bias towards certain functional categories. (c) How was the missing data handled in analysis? (d) Provide a supplementary summary table of proteins, peptides, spectral counts and MSMS counts for each patient in the two study cohort. (e) Provide a supplementary table for all proteins identified in the study. (f) Page 9, Line 254. From Figure 4b it appears that C1RL expression is negative correlated with HbA1c level in the CRPS group. Please clarify. (g) Page 11, Section 3.4 The selection of 1.1 fold change in conjunction with a simple p-value threshold of 0.05 appears very relaxed for any proteomics study? Please explain why a 10% change threshold was considered to be appropriate for this study (clinical information, relevant literature etc). A corrected p-value (q-value) could be employed or verify some of the proposed markers using an orthogonal approach. (h) Provide ProteomeXchange dataset identifier to locate the study. (i) Spelling corrections - Table 2 description (spectural), Figure 2 (inglammatory).
Round 2
Reviewer 1 Report
Authors still say "protein quantitative ratio measurement" on line 284 and other places where they were talking about LC-MS/MS results. LC-MS/MS isn't quantitative by any means. Authors need to correct this wherever applicable.
Reviewer 3 Report
The revised manuscript by Chang et al. fails to address some of the critical comments noted in initial review. Most of the changes are superficial and no new experiments or verification studies were conducted to verify the proposed biomarkers. In summary, the manuscript is not fit for publication in its current state.
(a) The method section still does not provide enough details to replicate the study. Details on immunodepletion method are missing. The addition of term sixplex labeled is not clear?
(b) The revised manuscript does not include any new experiments to verify the proposed biomarkers. Goal of a verification study is to confirm the findings using an orthogonal approach. Authors in this study selected ELISA which failed to verify majority of the proposed biomarkers. Considering this was a pilot study with very small sample set and very relaxed threshold for selection of candidate biomarkers, it becomes very important to confirm results using an alternative approach. The idea is not to pit ELISA vs MSMS. The results can be verified using targeted proteomics approach or validated in a different cohort. However, the authors have not provided any new data to validate/verify their proposed biomarkers.
(c) Authors cite a study by Sergio Oller Moreno et al (2018) for their selection of 10% threshold. However, the published study had a very large sample size (total of 473 samples) compared to 13 samples analyzed by Chang et al. In addition Moreno et al conducted adequate data analysis to justify their 10% threshold which has not been done in the current manuscript.
(d) The provided supplementary protein and peptide tables are not at all useful in evaluating results as the filenames do not include details on which study cohort they belong.
(e) The authors do not address the question on their selection of large uniprot database for searching mass spectrometry data.
Round 3
Reviewer 3 Report
The authors have addressed most of my comments. However, few minor edits are needed before publication.
(a) The supplementary data needs considerable improvement. Provide separate tables for protein and peptides.
(b) The study cohorts are still not clearly labeled in the sample identifier sheet. For e.g. "CR-B13_170901.raw --> Sample 1" doesn't inform to which study group this sample belongs to. Is it the CR cohort or CRPS study cohort. Also whether it is pre-intervention plasma sample or post-intervention plasma sample is not clear from the file/sample identifier.
(b)The abundance data reported in Table 2 could not be replicated from the provided supplementary table. Include an additional supplementary table for just the 6 proteins of interest. with spectral counts for each member in the two study cohorts and showing how the data in table 2 was generated.
(c) If the 25% threshold retained all the identified proteins of interest, then authors should replace volcano plots with increased threshold and modify the selection criteria appropriately in the body of the paper.
The manuscript can be considered for publication if these comments are addressed.
